# Machine learning models identify predictive features of patient mortality across dementia types

Jimmy Zhang [1,2,5], Luo Song [3,5], Zachary Miller[4], Kwun C. G. Chan[4] & Kuan-lin Huang [1✉]

## Abstract

**Background** Dementia care is challenging due to the divergent trajectories in disease progression and outcomes. Predictive models are needed to flag patients at risk of near-term mortality and identify factors contributing to mortality risk across different dementia types.

**Methods** Here, we developed machine-learning models predicting dementia patient mortality at four different survival thresholds using a dataset of 45,275 unique participants and 163,782 visit records from the U.S. National Alzheimer's Coordinating Center (NACC). We built multi-factorial XGBoost models using a small set of mortality predictors and conducted stratified analyses with dementiatype-specific models.

**Results** Our models achieved an area under the receiver operating characteristic curve (AUC-ROC) of over 0.82 utilizing nine parsimonious features for all 1-, 3-, 5-, and 10-year thresholds. The trained models mainly consisted of dementia-related predictors such as specific neuropsychological tests and were minimally affected by other age-related causes of death, e.g., stroke and cardiovascular conditions. Notably, stratified analyses revealed shared and distinct predictors of mortality across eight dementia types. Unsupervised clustering of mortality predictors grouped vascular dementia with depression and Lewy body dementia with frontotemporal lobar dementia.

**Conclusions** This study demonstrates the feasibility of flagging dementia patients at risk of mortality for personalized clinical management. Parsimonious machine-learning models can be used to predict dementia patient mortality with a limited set of clinical features, and dementiatype-specific models can be applied to heterogeneous dementia patient populations.

## Plain language summary

Dementia has emerged as a major cause of death in societies with increasingly aging populations. However, predicting the exact timing of death in dementia cases is challenging, due to variations in the gradual process where cognitive decline interferes with the body's normal functions. In our study, we build machine-learning models to predict whether a patient diagnosed with dementia will survive or die within 1, 3, 5, or 10 years. We found that the prediction models can work well across patients from different parts of the US and across patients with different types of dementia. The key predictive factor was the information that is already used to diagnose and stage dementia, such as the results of memory tests. Interestingly, broader risk factors related to other causes of death, such as heart conditions, were less significant for predicting death in dementia patients. The ability of these models to identify dementia patients at a heightened risk of mortality could aid clinical practices, potentially allowing for earlier interventions and tailored treatment strategies to improve patient outcomes.

[1] Department of Genetics and Genomic Sciences, Center for Transformative Disease Modeling, Tisch Cancer Institute, Icahn Institute for Data Science and Genomic Technology, Icahn School of Medicine at Mount Sinai, New York, NY 10029, USA. [2] Columbia University, New York, NY 10027, USA. [3] School of Medicine, The University of Queensland, Herston, QLD 4006, Australia. [4] National Alzheimer's Coordinating Center, University of Washington, Seattle, WA 98195, USA. [5] These authors contributed equally: Jimmy Zhang, Luo Song. ✉email: kuan-lin.huang@mssm.edu

Dementia has become a growing public health concern, classified as the seventh leading cause of death[1] and the fourth most burdensome disease or injury in the United States in 2016 based on years of life lost[2]. As of 2022, an estimated $1 trillion of global annual costs[3] can be attributed to Alzheimer's disease and other dementias, affecting an estimated 6.5 million Americans[4] and 57.4 million people worldwide, and those numbers are expected to triple by 2050[5]. Unfortunately, the true mortality burden associated with dementia may still be underestimated, as dementia itself tends to be underreported on death certificates as the underlying cause of death[6].

This immense healthcare burden of dementia can be attributed to the lack of curative drugs[7,8], the challenge in predicting patient trajectory, and the intrinsic difficulty in diagnosing dementia, which often requires the evaluation of various criteria[9,10], including risk factors (e.g., old age and family history), cognitive impairment screening questionnaires, neuropsychological testing (e.g., the Mini-Mental State Examination (MMSE)[11] and the Alzheimer's Disease Assessment Scale—Cognitive Subscale (ADAS-Cog)[12]), physical examination, biomarkers, and neuroimaging. To aid the detection, diagnosis, and treatment of Alzheimer's disease and related dementias, the National Institute of Aging (NIA) founded the National Alzheimer's Coordinating Center (NACC)[13] in 1999. Using existing protocols, standardized and multi-institutional databases were collected and built over the past decades, encompassing clinical records for tens of thousands of patients that may be used to develop predictive models[14].

While machine-learning models have been developed for the diagnosis or classification of dementia[4,15–20], they have rarely been applied to the prediction of near-term survival or mortality in dementia patients. Most dementia survival prediction studies utilize traditional nonparametric estimator and regression models, such as Kaplan–Meier estimator curves and Cox proportional hazards models[21,22], rather than advanced machine-learning models. However, due to the issues of high dimensionality, non-linearity, censoring, heterogeneity, and missingness present in dementia clinical data, machine learning can often provide more accurate predictions as compared to traditional statistical methods[23]. In the few published studies that utilized machine learning to predict dementia patient mortality[23–25], the models achieved reasonable performance. However, two of these studies were conducted within small cohorts (<2000 patients) derived from single health systems or geographical regions, and important predictors of dementia patient mortality varied between studies. Moreover, none of these studies differentiated among dementia types, which remains a crucial next step for the personalized treatment and management of dementia[26,27]. A systematic effort to build predictive models using a cross-institutional database encompassing multiple dementia types is required to resolve these open questions.

To resolve these challenges, we utilize the NACC database, the largest resource of its kind in the United States, to (1) develop robust machine-learning models for predicting dementia patient mortality across various time frames, (2) identify key predictors of mortality, and (3) demonstrate how these predictors differ across dementia subtypes. Our models provide a robust method of flagging dementia patients at risk of near-term death, achieving an area under the receiver operating characteristic curve (AUC-ROC) of over 0.82 for all of 1-, 3-, 5-, and 10-year thresholds while utilizing a set of only nine features, most of which consist of dementia-related predictors as opposed to more general age-related risk factors. Our models also highlight crucial differences between dementia types, grouping vascular dementia with depression, Lewy body dementia with frontotemporal lobar dementia, and Alzheimer's disease with other (unclassified) dementia, in addition to revealing shared and distinct predictors

of mortality among the various dementia types. Overall, our models can be used both with a limited set of clinical features and in the presence of heterogeneous dementia patient populations, which can contribute to the precision care of dementia.

## Methods

**Data sources**. Our study utilized longitudinal data taken from the National Alzheimer's Coordinating Center (NACC) database, following over 40,000 unique patients and spanning 39 past and present Alzheimer's disease centers (ADCs) across the United States[13]. The NACC currently maintains a large relational database comprised of numerous individual datasets and forms. When the ADC program was first established in 1984, ADCs primarily collected cross-sectional data as part of a Minimum Data Set (MDS), which contained limited demographic and clinical data from each patient's most recent visit. However, after the NACC was established in 1999, ADCs began collecting more extensive neuropathological data via the 2001 NP form, and then in 2005, a comprehensive longitudinal dataset known as the Uniform Data Set (UDS), replaced the MDS and became standardized across all ADCs[13,28]. In 2015, version 3 of the UDS was implemented and currently remains in use[13].

The Institutional Review Board (IRB) at the University of Washington authorizes the release and research of the NACC database[28]. Each contributing ADC is required to obtain informed consent from all participants and receive approval from its own individual IRB in order to submit data to the NACC. The specific dataset used in this study was obtained from the NACC by submitting a data request that was approved. The National Alzheimer's Coordinating Center data used herein were de-identified. All methods were carried out in accordance with relevant guidelines and regulations, including the NACC Data Use Agreement.

The raw, unprocessed dataset used for our study contained data from June 2005 up to the September 2021 data freeze, comprising 163,792 patient visits and 1,061 variables. These variables constituted a combination of demographic, comorbidity, neurological examination, clinical diagnosis, neuropathological, and genetic data that are linked to the NACC's Uniform Data Set (UDS). The variables used in this study and their corresponding descriptors are available in Supplementary Table 1.

**Survival analysis**. To gain preliminary insights into the relationship between dementia and patient survival/mortality, we first conducted a survival analysis using global clinical dementia rating (CDR) and dementia type as stratification variables, excluding dementia types with fewer than 100 patients. For our survival analysis, we built Kaplan–Meier estimator curves with the "survfit" function from the *survival*[29] R package. We used each unique patient's first visit as the starting point for tracking patient survival, and we calculated days of survival since the first visit based on (1) the time of death if the patient's death was recorded within the timespan of the dataset or (2) the expiration date of the dataset if the patient was still alive.

**Data cleaning**. All data cleaning was conducted in R v4.1.2 (R Foundation for Statistical Computing, Vienna, Austria). First, we preserved NACCID (subject ID number) and NACCADC (ADC at which the subject was seen) but removed all other form header information and text field variables. We then re-encoded the remaining features, which consisted primarily of categorical variables that were originally encoded as type numeric. Accordingly, we converted all "Not available" codes (−4 and −4.4), "Not assessed" or "Not applicable" codes (8, 88, 888, 888.8, and 8888), and "Unknown" codes (9, 99, 999, 999.9, 9999) to NA, accounting

for the specific special code(s) corresponding to each variable. Additionally, we re-encoded the extra categories in the Neuropsychological Battery Summary Score variables (95, 96, 97, 98, 995, 996, 997, 998), which corresponded to an inability or refusal to complete the Mini-Mental State Exam (MMSE), to NA as well. Any variables with inconsistent coding (e.g., RACE with code 50 for "Other") were manually re-encoded as appropriate. Finally, we converted all categorical variables from type numeric to type factor.

**Missing data imputation**. Generally, readily available machine-learning models are not compatible with missing data. Moreover, having large amounts of missing data can often affect model performance and generalizability across populations. Within the NACC dataset, which contains a large feature space and missing values scattered across variables, the removal of a row due to a single missing value can be especially detrimental and drastically reduce sample sizes. To avert potential bias introduced by manually selecting features, we opted to impute variables with missing values rather than only include patients with complete data.

The NACC Uniform Data Set has undergone several revisions since its inception in 2005, and the most recent version (version 3) was implemented in 2015. Consequently, certain variables that were collected in older versions of the UDS were no longer collected in UDS v3, and vice versa. Therefore, to minimize the number of features that did not contain sufficient non-missing values, we first omitted all variables with over 40% missing values. For the 189 remaining features, we imputed missing values using MICE (Multivariate Imputation by Chained Equations)[30]. Multiple imputation is an imputation strategy that accounts for variability in missingness by generating multiple imputed datasets, which can then be aggregated into a single complete dataset[31]. Thus, multiple imputation generally outperforms traditional machine-learning methods used for imputation[32,33]. MICE implements a form of multiple imputation that relies on predictive mean matching to predict the value of a given missing variable based on data points that most closely resemble the missing data point.

**Data splitting**. To evaluate patient survival status, we employed 1 year, 3 years, 5 years, and 10 years as survival-time thresholds. Accordingly, we determined each patient's survival status based on survival threshold year length, clinic visit date, and patient's time of death that was derived from variables NACCMOD (Month of Death) and NACCYOD (Year of Death), labeling each patient's 1-year, 3-year, 5-year, and 10-year survival as either 0 (survival) or 1 (deceased).

To assess the accuracy of our prediction models, we divided the whole cohort into a training/internal test dataset and a separate external test dataset. In addition, to maximize the utilization of the dataset, our train/internal-test datasets contained all patients who visited before June 2019—[Survival Years] or died before June 2019, while our external test datasets contained all patients who visited after June 2019—[Survival Years] and had not died by June 2019. Thus, the model would have no access to the new records from the external test set in the training phase. For all train/internal-test datasets and external test datasets, we firstly excluded patients with unknown time of death or lost follow-up (NACCMOD or NACCYOD = "99/9999: Unknown") and kept all non-deceased patient records (NACCMOD and NACCYOD = "88/8888: subjects not deceased") and patient records with a specific time of death. Then, we labeled patients who died before [Visit Date] + [Survival Years] as 1 (deceased), while the others in the datasets were labeled as 0 (survival).

Subsequently, for each survival-time threshold, we stratified each survival dataset by date into a pan-dementia dataset that we used for training and internal tests and a separate external test cohort that we used to externally evaluate model performance. For our pan-dementia analysis, we included all dementia patients (i.e., all patients who received an impaired, not MCI, MCI, or dementia diagnosis). However, for our sub-dementia analysis, we stratified our datasets by dementia type, including non-dementia patients as a baseline for comparison.

**Machine-learning models**. After experimenting with several machine-learning algorithms (i.e., random forest, logistic regression, and gradient boosting), our machine-learning algorithm of choice was eXtreme Gradient Boosting (XGBoost), a high-performance, tree-based ensemble learning method that uses gradient tree boosting to sequentially add new trees to reduce the errors from previous trees[34]. As compared to other gradient-boosting algorithms like light gradient-boosting machine (LightGBM), previous literature has found that XGBoost provides an optimal balance between accuracy and training speed[35]. When we evaluated the predictive performance of XGBoost vs. LightGBM, we found that they achieved AUC-ROC scores within 0.01 of each other at all four survival thresholds: 0.81 vs. 0.82 for 1-year, 0.82 vs 0.83 for 3-year, 0.82 vs. 0.83 for 5-year, and 0.83 vs. 0.83 for 10-year. Thus, we opted to proceed with XGBoost for the rest of our analyses. We built XGBoost models for each of the 1-year, 3-year, 5-year, and 10-year datasets, with the goal of predicting dementia patient survival/mortality under varying survival thresholds. We built all of our machine-learning models in Python v.3.7.12 using the *xgboost* and *scikit-learn*[36] libraries.

**Class imbalance**. All four survival datasets exhibited some degree of class imbalance: the 1-year, 3-year, and 5-year datasets contained a higher proportion of survival patients than mortality patients, whereas the 10-year dataset contained a higher proportion of mortality patients than survival patients. In order to address this class imbalance, we chose to apply a class-weighted loss function when training our models via XGBoost's built-in "scale_pos_weight" parameter. This parameter controls the balance of positive and negative weights, such that setting this parameter equal to the ratio between the number of samples in the negative class and the number of samples in the positive class produces a class-weighted loss function for the XGBoost model[37]. For the two-feature and multi-factorial models, the "scale_pos_weight" parameter was set to 15.081 for 1 year, 3.862 for 3 years, 1.872 for 5 years, and 0.476 for 10 years (Supplementary Table 2). We opted to use a class-weighted loss function rather than a class-based resampling method because previous studies have demonstrated the superior performance of weighted methods as compared to resampling methods in addressing class imbalance[38], and we also wanted to avoid the additional computational complexity incurred by oversampling.

**Feature selection**. For our pan-dementia analyses, we aimed to build XGBoost models to predict 1-year, 3-year, 5-year, and 10-year survival among all dementia patients. Our first set of machine-learning models utilized only two features: age and standard global CDR. These preliminary models served as a baseline of comparison for the more complex models and provided insight into how much of the mortality prediction could be explained by age and standard global CDR alone.

Subsequently, we built a more complex set of machine-learning models that employed the larger feature space. However, in order to make our machine-learning models more clinically feasible, we conducted feature selection using SHapley Additive exPlanations

(SHAP)[39], a unified, model-agnostic framework for interpreting the predictions of machine-learning models. The SHAP algorithm is rooted in game theory, relying on the calculation of Shapley values to evaluate the relative contribution of each feature to a given prediction. Though SHAP is most often used as a feature importance metric, it has demonstrated considerable utility as a feature selection method as well, even outperforming many conventional feature selection methods[40]. In our study, we trained default XGBoost classifiers with five-fold cross-validation on each of the four training sets, aggregating SHAP values across each partition before taking the union of the top five features from each model, ranked in order of decreasing mean absolute SHAP value.

**Model training, internal testing, and external testing**. We trained our four XGBoost models on their respective training sets and tested their performance on their respective internal test sets, corresponding to their survival threshold. To account for any variability that may have been introduced by the random state of the train-internal test split, we conducted bootstrap resampling by generating fifty bootstrap samples, re-fitting the models on each bootstrap train set, and evaluating their performance on each bootstrap test set. All confidence intervals generated represent the 95% confidence intervals derived from bootstrap resampling. We also validated each model's performance on its respective external test set, which we set aside during data splitting.

**Hyperparameter optimization**. To optimize model performance, we used Bayesian optimization to identify the optimal hyperparameters for each XGBoost model, implemented with the *BayesianOptimization*[41] Python library. Bayesian optimization is a robust hyperparameter optimization algorithm that employs Bayes' theorem and Gaussian processes to efficiently search the hyperparameter space. Given a black-box function, Bayesian optimization builds a probabilistic surrogate model of the objective function, which is then searched by an acquisition function that incrementally selects hyperparameters to optimize the surrogate model[42,43].

For each model, we applied fifty rounds of Bayesian optimization with five-fold cross-validation, optimizing the following hyperparameters: "n_estimators", "max_depth", "colsample_bytree", "min_child_weight", "learning_rate", "subsample", and "gamma". Additionally, to account for class imbalances, we set the "scale_pos_weight" parameter of each model to the ratio between the number of samples in the negative class (survival) and the number of samples in the positive class (mortality). The full lists of tuned hyperparameters for our two-feature, multi-factorial, and dementia type-specific models are available in Supplementary Table 2.

**Sub-dementia analysis**. In addition to predicting survival in all dementia patients, we conducted a sub-dementia analysis, analyzing discrepancies among dementia types. Since the majority of dementia-related studies are geared toward Alzheimer's disease, highlighting the distinctions between dementia types may provide insight into the mechanisms of the various forms of neurodegeneration, thus guiding clinical practice.

For our sub-dementia analysis, we only used a 5-year survival threshold, as the pan-dementia analysis demonstrated that 5 years provides a reasonable timeframe for capturing patient mortality without a drastic trade-off in predictive performance. To ensure that each of our dementia type-specific models received sufficient training data, we limited our analysis to eight dementia types, which each contained at least 1000 patients from the 5-year dataset between training and internal test (excluding external

test). Accordingly, we built XGBoost models for each sub-dementia dataset and applied the same Bayesian optimization methodology and train/internal-test/external-test framework as with our pan-dementia analysis. However, in order to conclusively note differences between dementia types, we included all 189 original features and allowed each model to designate the most important features corresponding to its respective dementia type.

**Feature importance**. For both our pan-dementia analysis and sub-dementia analysis, we used the aforementioned SHapley Additive exPlanations (SHAP)[39] to determine feature importance within our XGBoost models. We used a variant of SHAP known as TreeSHAP[44], an enhancement to SHAP designed for tree ensemble methods, such as XGBoost. To distinguish the most important features in each model, we created 50 bootstrap samples with randomized train-internal test configurations, fit the model on each training split, and then calculated SHAP values within each internal test split. We then aggregated the SHAP values across all bootstrap samples before ranking the features in order of decreasing mean absolute SHAP value, based on their relative contribution to the models.

**Reporting summary**. Further information on research design is available in the Nature Portfolio Reporting Summary linked to this article.

## Results

**Cohort characteristics and patient mortality across dementia types**. Data for this study was obtained from the National Alzheimer's Coordinating Center (NACC) database[13], which spans 39 past and present Alzheimer's Disease Centers (ADCs) across the United States. The NACC collects, audits, and distributes ADC-derived data across the U.S. The NACC data release used for this study included 45,275 unique participants and 163,782 visit records between 2005 and 2021.

Data were extracted for various dementia severity levels (Table 1 and Supplementary Table 3). The mean age at visit increased from 73 years for those who had normal cognition ($n = 16,379$), to 76 years for those with dementia ($n = 19,186$). The percentage of females was 64.9% among normal cognition patients, compared to 52.0% among demented patients. For impaired-not-MCI ($n = 1840$), MCI ($n = 7870$), and dementia groups, the proportions of females were 58.3%, 52.4%, and 52.0%, respectively. Mean education years were 15–16 across all dementia severity levels in this population. The percent of participants with at least one AD risk APOE e4 allele was 23.1% among normal cognition participants, compared to 38.5% among demented participants. This NACC cohort was utilized for statistical analyses and machine-learning model training in this work (Fig. 1A).

The whole cohort with 45,275 unique NACC individuals in this 2005–2021 time period was analyzed to compare patient survival across different dementia types. We estimated survival time using the Kaplan–Meier method (Fig. 1B, C). Survival probability differed across primary etiologic diagnoses of dementia types (Fig. 1B). Patients with prion disease showed less overall median survival time than other dementia types ($p < 0.0001$). This is consistent with the rapid onset and progression feature of prion disease[45]. The overall median survival time for Alzheimer's disease was not reached, with 5- and 7-year survival rates of 76.05% and 66.63%, respectively. The overall median survival time in Lewy Body disease was 98.3 months (95% CI 84.2–119.5), with 5- and 7-year survival rates of 60.0% and 52.0%, respectively.

**Table 1 Characteristics of NACC participants by cognitive status.**

| Characteristic | Dementia severity | | | |
| --- | --- | --- | --- | --- |
| | Normal cognition | Impaired-not-MCI | MCI | Dementia |
| Number of participants | 16379 | 1840 | 7870 | 19186 |
| Age (years old), mean (SD) | 73 (12) | 73 (11) | 76 (10) | 76 (11) |
| Female, n (%) | 10632 (64.9%) | 1071 (58.3%) | 4124 (52.4%) | 9976 (52.0%) |
| Education (years), mean (SD) | 16 (7) | 15 (6) | 16 (8) | 16 (9) |
| Race, n (%) | | | | |
| White | 12869 (78.6%) | 1335 (72.6%) | 6048 (76.8%) | 16169 (84.3%) |
| Black/African American | 2629 (16.1%) | 348 (18.9%) | 1324 (16.8%) | 1918 (10.0%) |
| American Indian/Alaskan Native | 138 (0.8%) | 23 (1.3%) | 83 (1.1%) | 151 (0.8%) |
| Native Hawaiian/Pacific Islander | 14 (0.1%) | 5 (0.3%) | 5 (0.1%) | 25 (0.1%) |
| Asian | 492 (3.0%) | 40 (2.2%) | 234 (3.0%) | 420 (2.2%) |
| Other/multiracial/unknown | 237 (1.4%) | 89 (4.8%) | 176 (2.2%) | 503 (2.6%) |
| Hispanic ethnicity, n (%) | 1115 (6.8%) | 234 (12.7%) | 766 (9.7%) | 1473 (7.7%) |
| >= 1 APOE e4 allele, n (%) | 3781 (23.1%) | 400 (21.7%) | 2152 (27.3%) | 7395 (38.5%) |

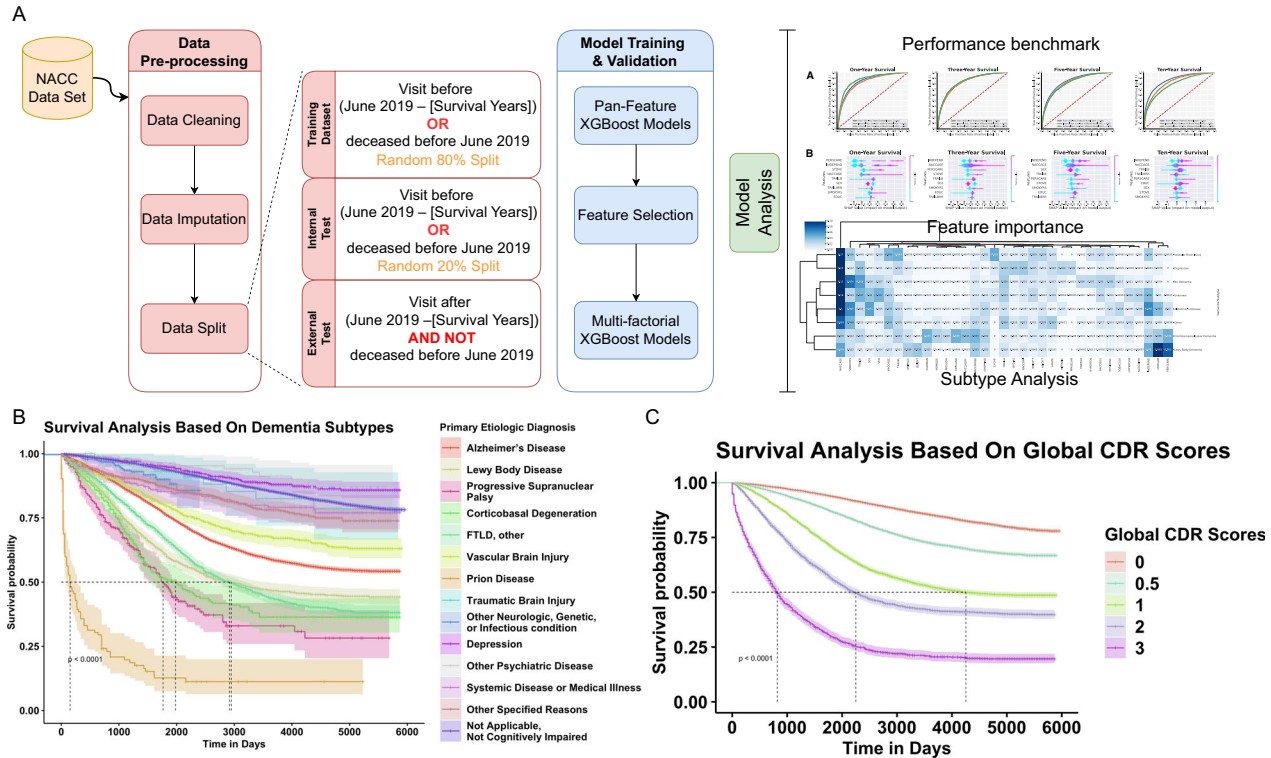

**Fig. 1 Schematic diagram of the study workflow and survival analyses using the NACC dataset. A** Depiction of the study workflow from data pre-processing, model training, and testing, to analyses of the machine-learning models. The survival analysis results in the NACC datasets are visualized by the Kaplan–Meier estimator curves separated by **B** dementia type and **C** standard global clinical dementia rating (CDR), where the shaded area around each line represents a 95% pointwise confidence interval.

To illustrate the relationship between dementia severity and survival, we performed a survival analysis based on global clinical dementia rating (CDR) scores (Fig. 1C). The overall median survival time in global CDR score at 0 was not reached, with 5- and 7-year survival rates of 93.7% and 90.1%, respectively. The overall median survival time in global CDR scores at 1, 2, and 3 was 141.8 months (95% CI 128.8–NA), 75.0 months (95% CI 69.0–81.0), 27.3 months (95% CI 25.3–29.3), respectively. With increasing global CDR score, which represents more severe cognitive impairment, patients generally showed worse outcomes. Moreover, to determine whether these trends were reflected in patients with comorbidities (i.e., other causes of death such as cancer and cardiovascular disease), we performed additional

survival analysis on global CDR scores stratified by disease type. The results showed that regardless of whether or not dementia patients suffered from cancer or heart conditions, global CDR score remained significantly associated with mortality (Supplementary Fig. 1), suggesting that—even among dementia patients with comorbidities—dementia-related causes are likely still the dominating factors of mortality.

**Predicting dementia patient mortality using age and standard global CDR.** Since our survival analysis revealed that higher standard global CDR coincides with a faster decline in survival probability, we first built simplistic eXtreme Gradient Boosting

**Table 2 Predictive performance of the nine-feature, multi-factorial models.**

| Survival-time threshold | Internal test set | | | External test set | | |
|---|---|---|---|---|---|---|
| | Accuracy (95% CI) | AUC-ROC (95% CI) | AUC-PR (95% CI) | Accuracy | AUC-ROC | AUC-PR |
| One-year | 0.780 (0.770–0.788) | 0.824 (0.820–0.850) | 0.259 (0.257–0.301) | 0.817 | 0.870 | 0.300 |
| Three-year | 0.750 (0.743–0.758) | 0.825 (0.817–0.830) | 0.566 (0.545–0.588) | 0.752 | 0.837 | 0.539 |
| Five-year | 0.744 (0.735–0.749) | 0.823 (0.813–0.826) | 0.722 (0.702–0.730) | 0.710 | 0.817 | 0.667 |
| Ten-year | 0.748 (0.733–0.755) | 0.829 (0.814–0.832) | 0.905 (0.896–0.911) | 0.693 | 0.789 | 0.741 |

(XGBoost) ML models that utilize only two features, age, and standard global CDR, to predict mortality in dementia patients. We stratified our data into four datasets separated by survival endpoints, each with an 80/20 train/internal-test split and a separate external test cohort of later visits not seen by the model: 1-year survival (train: $n = 60{,}367$; internal test: $n = 15{,}092$; external test: $n = 10{,}284$ visit records), 3-year survival (train: $n = 53{,}272$; internal test: $n = 13{,}318$; external test: $n = 11{,}552$), 5-year survival (train: $n = 47{,}196$; internal test: $n = 11{,}800$; external test: $n = 11{,}284$), and 10-year survival (train: $n = 32{,}569$; internal test: $n = 8143$; external test: $n = 13{,}266$) (Supplementary Fig. 2). We trained our models accordingly and employed Bayesian optimization[41] to select the optimal hyperparameters for each model.

The two-feature XGBoost models achieved an AUC-ROC of over 0.76 at all survival thresholds, though the higher thresholds achieved much higher AUC-PR (area under the precision-recall curve) scores, likely due to the large class imbalances at the lower thresholds. The full table of model performance for the two-feature XGBoost models in the internal and external test sets is shown in Supplementary Table 4, and the AUC-ROC curves are shown in Supplementary Fig. 3. Overall, these basic models confirmed that age and clinical dementia rating alone provide a reasonable prediction of dementia patient mortality, and their contributions would be further elucidated with the inclusion of more clinical features in our subsequent analyses.

**Multi-factorial machine-learning models for predicting mortality in dementia patients.** We proceeded to build multi-factorial models that introduced a wider array of clinical features into the machine-learning models. Initially, we built XGBoost models encompassing all 189 features of our preprocessed datasets. We then used SHapley Additive eXplanations (SHAP)[40], a robust, game-theoretic framework for explaining model output, to identify the important features within each survival threshold. These initial results identified numerous recurring features among the top features, ranked by SHAP values, of each of the four survival-time thresholds, and much of the explainability of the predictions could be attributed to these top few features alone (Supplementary Fig. 4a).

Therefore, we derived a parsimonious and informative feature subset across all four survival-time thresholds by training default XGBoost classifiers with five-fold cross-validation on each of the four training sets and taking the union of the top $n$ features from each model. Specifically, we selected $n = 5$ as our cutoff as it provided considerable overlap among survival thresholds while still preserving enough features to be clinically informative (Supplementary Fig. 4b, c). Nonetheless, the selection criteria constitute a deliberate compromise, striking a balance between optimizing model performance and ensuring practicality for clinical applications.

The SHAP bar plots showing all features at each survival threshold are shown in Supplementary Fig. 4a, where we retain the top features. Notably, other leading causes of death in the US,

such as stroke and other cardiovascular conditions, were ranked outside of the top 20 features at all survival thresholds.

After conducting feature selection, our resulting feature subset consisted of nine parsimonious features: "NACCAGE" (Subject's age at visit), "INDEPEND" (Level of independence), "PERS-CARE" (Personal care), "TRAILB" (Trail Making Test Part B—Total number of seconds to complete), "STOVE" (In the past four weeks, did the subject have any difficulty or need help with Heating water, making a cup of coffee, turning off the stove), "SEX" (Subject's sex), "SMOKYRS" (Total years smoked cigarettes), "TRAILBRR" (Trail Making Test Part B—Number of commission errors), and "EDUC" (Years of education).

Utilizing the same train/internal-test splits and external test cohorts as in the two-feature models, we employed this subset of nine features on new, Bayesian-optimized XGBoost models to predict survival/mortality across all dementia patients at each survival threshold. All four models achieved an AUC-ROC of over 0.82, though the lower threshold models (i.e., 1-year and 3-year) struggled with respect to AUC-PR. At the 10-year survival threshold, our model achieved the highest AUC-ROC and AUC-PR of all, with an AUC-ROC of 0.829 (95% CI: 0.814–0.832) and an AUC-PR of 0.905 (95% CI: 0.896–0.911). Notably, model AUC-PR was worse in 1/3-year survival but increased dramatically at higher survival-time thresholds, which can likely be attributed to a higher proportion of mortality in patients and, thus, smaller class imbalances at these higher thresholds. Whereas the AUC-ROC metric can be optimistic on imbalanced classification problems, which is exemplified by the relative consistency of AUC-ROC scores across all four survival thresholds, AUC-PR is more strongly affected by class imbalance, which is likely why AUC-PR scores improved so dramatically as the proportion of mortality patients increased. Moreover, these performance trends were reflected in the external test sets as well. The full table of model performance for the multi-factorial models in the internal and external test sets is shown in Table 2, and the AUC-ROC curves are shown in Fig. 2A.

Additionally, to determine whether model performance was consistent across Alzheimer's disease centers (ADCs), we verified the performance of each model across all ADCs with at least 200 patients in the internal test set at each survival threshold. To avoid potential data leakage between sites, we performed leave-one-(site)-out cross-validation (LOOCV), in which training data from all but one site was used for training, while the internal test data from the leftover site was used for testing. This way, each ADC was tested and evaluated using models that were trained on data exclusively from other ADCs. Overall, model performance remained consistent across ADCs, with discrepancies primarily occurring in the ADCs with the smallest patient populations. The full AUC-ROC curves stratified by ADC are available in Supplementary Fig. 5 and demonstrate the broad generalizability of our model.

We generated the bootstrapped SHAP plots (Fig. 2B) of the multi-factorial models to reveal key insights about the nine chosen features in relation to dementia patient mortality. Another key strength of the SHAP method is its ability to determine not

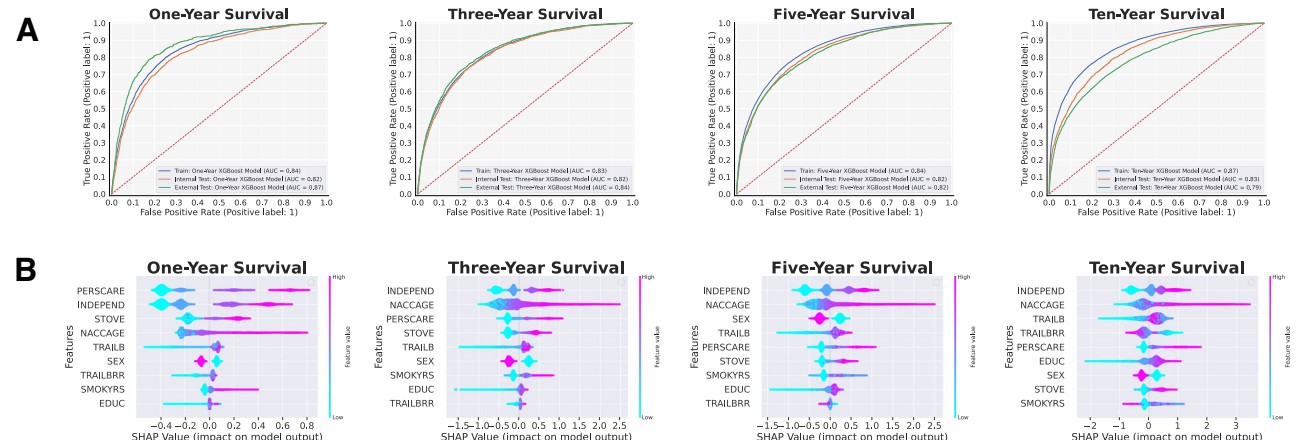

**Fig. 2 Receiver operating characteristic (ROC) curves and feature importance plots of the multi-factorial models predicting survival of NACC individuals at different time points. A** ROC curves of the 1-year, 3-year, 5-year, and 10-year survival models. **B** Feature importance as illustrated by the SHapley Additive exPlanations (SHAP) beeswarm plots of the 1-year, 3-year, 5-year, and 10-year survival models. The SHAP values on the x-axis have log-odds units and represent the impact of the feature on the model output. For each feature, each individual patient is represented by a single point, and the x-position of a point represents the impact of that feature on that patient. The color of a point corresponds to the patient's value for that feature along a continuous scale from low to high.

only the magnitude of a feature's effect on model prediction but also the direction of its effect. Notably, a higher risk of mortality (positive SHAP value) was predicted by old age ("NACCAGE"), male sex ("SEX"), higher levels of dependency ("INDEPEND"), higher levels of personal care required ("PERSCARE"), greater difficulty in handling a stove or heating water ("STOVE"), more years of education ("EDUC"), and more seconds required to complete the Trail Making Test Part B ("TRAILB") across all four survival thresholds. More years of smoking ("SMOKYRS") and more commission errors on the Trail Making Test Part B ("TRAILBRR") were also predictive of higher risk in 1-year and 3-year mortality. However, the inverse relationship was observed for 5-year and 10-year mortality risk, as fewer years of smoking and fewer commission errors on the Trail Making Test Part B became slightly more predictive of mortality than of survival at these longer survival thresholds. The full SHAP beeswarm plots for the multi-factorial models are shown in Fig. 2B.

**Dementia type-specific models**. The multi-factorial machine-learning models provided a cogent framework for predicting mortality in an unspecified population of dementia patients. However, across dementia types, there may have been key similarities and distinctions that could not be captured in the pan-dementia analysis. Therefore, we stratified the NACC cohort into smaller cohorts based on dementia types to conduct sub-dementia analyses. For these analyses, we aimed to predict dementia patient mortality solely at the 5-year survival threshold, which exhibits the smallest class imbalance and provides an extended time window for possible clinical actionability. We conducted our sub-dementia analysis on these eight dementia types with sample sizes greater than 1000: no dementia (n = 42,135 visit records), Alzheimer's disease (AD, n = 37,990), unknown (n = 6317), frontotemporal lobar degeneration (FTLD, n = 4290), Lewy body dementia (LBD, n = 3182), vascular brain injury or vascular dementia (VaD, n = 2288), cognitive impairment due to other reasons (n = 1362), and depression (n = 1354). We stratified the training, internal test, and external test sets of the 5-year dataset by dementia type and then trained, optimized, tested, and validated an XGBoost model for each of the eight dementia types.

Performance-wise, the models built on the commonly defined dementia types (e.g., AD, FTLD, LBD, and VaD) tended to

perform better in the positive class (mortality) and, thus, generally had higher AUC-PR, whereas the models built on the non-dementia patients and the more ambiguous dementia types (e.g., depression, cognitive impairment for other specified reasons, and missing/unknown) were more robust at predicting the larger negative class (survival) and, thus, had higher AUC-ROC overall. All eight models achieved an AUC-ROC of over 0.79, with the no-dementia model attaining the highest AUC-ROC at 0.873 (95% CI: 0.859–0.879). The most consistent, all-around performer was the AD model, which is reasonable given that it was by far the most popular dementia type aside from the no dementia group. The full table of model performance for the sub-dementia models in their respective internal and external test sets is shown in Table 3.

The clustered feature importance heatmap is shown in Fig. 3A, and the SHAP beeswarm plots for the dementia type-specific models are shown in Fig. 3B. Hierarchical clustering produced the following four clusters of dementia types: (1) VaD and depression, (2) FTLD and LBD, (3) AD and other dementia, and (4) no dementia and unknown. Notably, many of the key features in the pan-dementia cohort reappeared among the top features within most dementia types, including "NACCAGE" (Subject's age at visit), "INDEPEND" (Level of independence), and "SMOKYRS" (Total years smoked cigarettes). New features such as 'NACCADC' (ADC at which subject was seen), "VEG" (Vegetables—Total number of vegetables named in 60 s), "TRAVEL" (In the past four weeks, did the subject have any difficulty or need help with Traveling out of the neighborhood, driving, or arranging to take public transportation), and "TRAILA" (Trail Making Test Part A—Total number of seconds to complete) also emerged as important features across numerous dementia types.

Meanwhile, several key differences distinguished individual dementia types and their clusters from one another. For instance, in both VaD and depression, alongside general cognitive features, body measurements and vital signs, such as "HEIGHT" (Subject's height (inches)), "WEIGHT" (Subject's weight (lbs)), "NACCBMI" (Body mass index (BMI)), "HRATE" (Subject's resting heart rate (pulse)), and "BPDIAS" (Subject's blood pressure (sitting), diastolic), were more important for predicting mortality than for any other dementia type. In the vascular dementia subgroup, "CVCHF" (Congestive heart failure) was also

**Table 3 Predictive performance of the individual dementia type-specific models.**

| Dementia type (sample size [n] in internal test set/external test set) | Internal test set | | | External test set | | |
|---|---|---|---|---|---|---|
| | Accuracy (95% CI) | AUC-ROC (95% CI) | AUC-PR (95% CI) | Accuracy | AUC-ROC | AUC-PR |
| No dementia (n = 8427/11800) | 0.834 (0.827–0.844) | 0.873 (0.859–0.879) | 0.513 (0.475–0.546) | 0.696 | 0.842 | 0.336 |
| Alzheimer's disease (n = 7598/7351) | 0.774 (0.766–0.783) | 0.854 (0.845–0.863) | 0.790 (0.769–0.805) | 0.683 | 0.827 | 0.695 |
| Missing/unknown (n = 1264/191) | 0.808 (0.773–0.834) | 0.862 (0.813–0.889) | 0.504 (0.477–0.629) | 0.838 | 0.794 | 0.263 |
| Frontotemporal lobar degeneration (n = 858/1058) | 0.714 (0.683–0.745) | 0.796 (0.760–0.821) | 0.810 (0.735–0.819) | 0.695 | 0.772 | 0.677 |
| Lewy body dementia (n = 637/561) | 0.719 (0.696–0.769) | 0.796 (0.780–0.842) | 0.806 (0.771–0.846) | 0.717 | 0.807 | 0.763 |
| Vascular brain injury or vascular dementia (n = 458/507) | 0.751 (0.727–0.803) | 0.839 (0.796–0.871) | 0.752 (0.681–0.815) | 0.712 | 0.797 | 0.640 |
| Cognitive impairment for other specified reasons (n = 273/495) | 0.780 (0.744–0.828) | 0.832 (0.795–0.900) | 0.665 (0.469–0.731) | 0.786 | 0.833 | 0.421 |
| Depression (n = 271/235) | 0.815 (0.790–0.889) | 0.800 (0.732–0.895) | 0.408 (0.228–0.606) | 0.813 | 0.785 | 0.421 |

a pivotal feature, second in importance after age and accounting for over 5% of the mortality prediction among VaD patients.

In the FTLD and LBD cluster, feature importance was distributed across a substantially wider array of cognitive features, with less importance attributed to age and smoking years as compared to the other dementia types. In FTLD, for instance, a number of new cognitive features emerged: CDR® Plus NACC FTLD features (e.g., "CDRLANG" (Language) and "COMMUN" (Community Affairs)), clinician judgment features regarding motor function (e.g., "NACCMOTF" (Indicate the predominant symptom that was first recognized as a decline in the subject's motor function) and "MOMODE" (Mode of onset of motor symptoms)), and neuropsychological battery summary scores (e.g., "NACCMMSE" (Total MMSE score (using D-L-R-O-W))). Accordingly, difficulty in performing functional and social activities, in addition to the loss of motor function, were crucial predictors of mortality in FTLD patients, more so than in any other dementia type. As for LBD, "CDRSUM" (Standard CDR sum of boxes) superseded age as the most important feature, accounting for nearly 10% of the mortality prediction among LBD patients. However, other new features such as "ANIMALS" (Animals—Total number of animals named in 60 s), "ORIENT" (Orientation), and "NACCMMSE" (Total MMSE score (using D-L-R-O-W) were also revealed to be relevant to the mortality prediction within the LBD subgroup.

The top features in the AD subgroup comprised almost entirely cognitive features, most of which overlapped with those of the multi-factorial models, with the addition of "CDRSUM" (Standard CDR sum of boxes), "SHOPPING" (In the past four weeks, did the subject have any difficulty or need help with Shopping alone for clothes, household necessities, or groceries), and "TOBAC100" (Smoked more than 100 cigarettes in life). The top features in the other subgroup similarly consisted primarily of cognitive features, with the addition of body measurements and vital signs, similar to the VaD and depression cluster.

Finally, in the no dementia and unknown subgroups, many of the features that were typically associated with mental cognition, such as age and performance on neuropsychological exams, remained important predictors of mortality, though others were superseded by more general comorbidities and risk factors. For instance, the relative importance of "SMOKYRS" (Total years smoked cigarettes) was higher in the no dementia group than in any of the dementia groups, accounting for 7.5% of the mortality prediction among no dementia patients. Other non-cognitive risk factors such as "HYPERTEN" (hypertension) and "ENERGY" (Do you feel full of energy?) were also revealed to be relevant for predicting mortality in non-dementia patients, despite having little to no contribution to the predictions in the dementia groups. These results reaffirmed that mortality predictors differ between non-demented and dementia patients, who show multiple survival factors related to their neuropsychological ability.

Overall, in the context of VaD and depression, mortality prediction was predominantly determined by age, body measurements, and vital signs, which is not surprising given that patients with VaD and/or depression often present strong physical deficiencies, in addition to cognitive ones. Meanwhile, given that FTLD and LBD are both associated with alterations in personality, behavioral changes, and motor symptoms, it is plausible that the model would group these conditions together. As for patients with AD or other forms of dementia, mortality prediction was determined almost entirely based on age and cognitive features (e.g., performance on cognitive screening tests). We postulate that the grouping of AD with other forms of dementia may be attributed to the National Institute on Aging and Alzheimer's Association's (NIA-AA) diagnostic criteria for Alzheimer's disease, which relies on a process of exclusion.

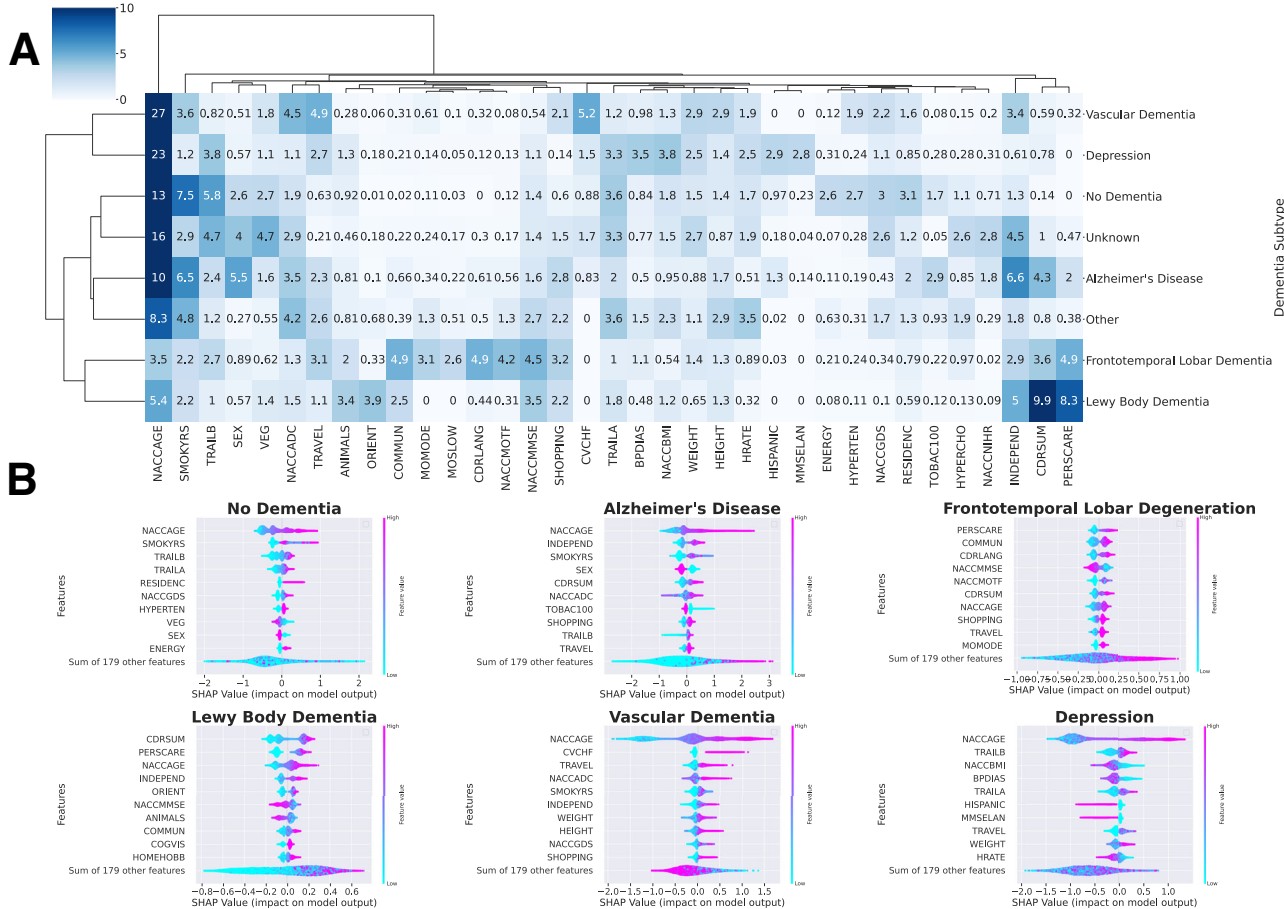

**Fig. 3 Comparison of features predictive of patient mortality across individual dementia type-specific models. A** Clustered heatmap of the top features across dementia types. Only features with a normalized SHapley Additive exPlanations (SHAP) value greater than 2.5 (explaining at least 2.5% of the prediction) in any given dementia type-specific model were included. The maximum normalized SHAP value for the clustered heatmap was set to 10 so that color contrasts were more discernable. The "No Dementia" category corresponds to patients receiving a primary etiologic diagnosis of "Not applicable, not cognitively impaired." The "Unknown" category corresponds to patients receiving a primary etiologic diagnosis of "Missing/unknown." "The "Other" category corresponds to patients receiving a primary etiologic diagnosis of "Cognitive impairment for other specified reasons (i.e., written-in values)." **B** SHAP beeswarm plots of six of the eight dementia type-specific models, excluding "Unknown" and "Other". Features in the beeswarm plots are ranked by mean absolute SHAP value. The SHAP value distribution of the top ten most important features in each dementia type-specific model is displayed.

Finally, the mortality prediction of individuals without dementia or with unknown dementia status was moderately influenced by age, biological sex, cognitive features, lifestyle factors (e.g., cigarette smoking), and geographical location. These findings align with previous research conducted on the general population.

## Discussion

In this study, we developed machine-learning models for predicting mortality through training, internal test, and external test sets using 163,782 visit records of 45,275 unique NACC individuals in the United States from 2005 to 2021. We have demonstrated that machine-learning models, which have thus far primarily been explored as screening or diagnosis tools in the context of dementia, have substantial utility in the prediction of mortality among dementia patients. First, we conducted multiple survival analyses, which confirmed that increasing global CDR scores coincided with decreased survival and showed that there was considerable variability in survival across dementia subtypes. Subsequently, we developed two-feature models (using only age and standard global CDR) and multi-factorial models (using nine features determined through feature selection) to predict dementia patient mortality at four distinct survival-time thresholds, all of which achieved high

predictive performance. We additionally built machine-learning models for eight different dementia subtypes and revealed key feature differences among them, though age and cognitive features derived from neuropsychological tests remained important predictors of mortality across all dementia types. These mortality predictors reveal similarities and differences in the etiology and clinical representation among individuals affected by different types of dementia.

The results of our global CDR survival analysis were consistent with those of past survival analyses in dementia patients[46,47], confirming that higher CDR scores correlate with reduced survival probability. With respect to dementia type, there have been very few studies investigating the association between dementia type and survival probability. In the studies that we identified, key distinctions were noted in the cohort composition and mortality risk of Lewy body dementia vs. Alzheimer's disease[27], vascular dementia vs. Alzheimer's disease in the context of depression[26], and among eight different dementia subtypes[48]. Our dementia-type survival analysis confirmed this heterogeneity, as survival probability differed drastically across groups of patients with different primary etiologic diagnoses. However, whereas prior studies identified comorbidities such as cardiovascular disease[49] to be associated with reduced survival probability, we found that

regardless of heart conditions, the survival curves separated decisively across patients with varied global CDR scores within the NACC cohort.

Subsequently, we built machine-learning models tasked with predicting dementia patient mortality at 1-, 3-, 5-, and 10-year survival thresholds. Our two-feature models, which utilized age and global CDR scores, achieved an AUC-ROC of over 0.76 at all four survival thresholds in the internal test set. Thus, age and global CDR provided a solid basis for predicting dementia patient mortality and, in the absence of additional clinical features, may alone be used to guide clinical judgment. Our multi-factorial models, for which we utilized SHAP to select a subset of nine features, achieved an AUC-ROC of over 0.82 at all four survival thresholds in the internal test set and comparable performance in the external test set. The crucial features used by the multi-factorial models confirm the known clinical indicators of dementia from a machine-learning standpoint. The multi-factorial models revealed that a higher risk of mortality was predicted by older age[50–53], male sex[46,51–53], higher levels of dependency and personal care required[53], more years of education[54], more years of smoking[55], and poorer performance on neuropsychological exams like the Trail Making Test[56,57]. Interestingly, at the longer survival thresholds, more years of smoking and more commission errors on the Trail Making Test Part B became more associated with survival than with mortality. Some possible explanations for these trends are that any protective effects of smoking cessation (which was not captured by the "SMOKYRS" variable) against dementia risk and mortality might only be realized in long-term outcomes[58] and that the amount of time required to complete the Trail Making Test Part B (captured by "TRAILB") might be a more consistent performance metric for predicting long-term mortality than the number of commission errors made (captured by "TRAILBRR"). However, it is also important to note that the data for the longer survival thresholds included a slightly different subset of patients (i.e., a higher proportion of patients from the early 2010s) due to the more years of follow-up data required, so future studies might seek to further investigate these hypotheses.

To our knowledge, our study is one of just a few studies to apply a machine-learning-based approach to predicting mortality in dementia patients[23–25] (as opposed to statistical approaches), and the first study to do so within population subsets stratified by dementia type. In predicting dementia patient mortality at the 5-year survival threshold, our dementia type-specific models all achieved an AUC-ROC of over 0.79 in the internal test set and similar performance in the external test set. Hierarchical clustering of survival predictors grouped the following dementia types together: (1) vascular dementia (VaD) with depression, (2) Lewy body dementia (LBD) with frontotemporal lobar dementia (FTLD), (3) Alzheimer's disease (AD) with other dementia, and (4) no dementia with unknown. Since many dementia types present similar symptoms and disease progressions[8], differentiating and targeting dementia type-specific symptoms and mortality predictors can be beneficial for patient populations[59]. Across all four clusters (even in the no dementia and unknown cluster), many features from the multi-factorial models remained key predictors of mortality, such as age, level of independence, smoking, and performance on neuropsychological exams like the Trail Making Test.

First, within the VaD and depression cluster, body measurements and vital signs (e.g., height, weight, BMI, heart rate, and diastolic blood pressure) contributed to the mortality prediction more than for any other dementia type. For VaD, congestive heart failure was the second most important feature after age, consistent with VaD's common risk factors[8]. Moreover, the grouping of VaD with depression confirms previous literature that has highlighted the synergistic effects of VaD and depression

on patient mortality[26], as VaD patients tend to exhibit a higher baseline risk for psychiatric symptoms like depression[59,60]. Second, within the FTLD and LBD cluster, features corresponding to MMSE score, standard CDR sum of boxes, and involvement in community affairs contributed more heavily to the mortality prediction. For FTLD in particular, features measuring difficulty in performing social and functional activities were the pivotal predictors of mortality, consistent with the pathological effects of FTLD[8]. Our findings regarding FTLD and LBD align with prior studies that have similarly grouped the two subtypes together and determined that executive dysfunction and activity disturbances are the key indicators of cognitive impairment for both[59,61]. Third, within the AD and other dementia cluster, general cognitive features, namely those from the multi-factorial models, remained the most important predictors of mortality. Standard CDR sum of boxes was also an important predictor of mortality in AD patients, as were body measurements and vital signs for other dementia patients. The grouping of AD with other dementia may be attributed to the difficulty in differentiating AD from certain other types of dementia[62], and given that AD was by far the most prevalent dementia type in the NACC cohort, it is likely that the other dementia patients were generally similar to AD patients. Finally, within the no dementia and unknown cluster, general cognitive features such as performance on the Trail Making Test, surprisingly, remained important predictors of mortality. However, general comorbidities and mortality risk factors, such as smoking, hypertension, and lack of energy, demonstrated high relative importance as well, more so than for any of the dementias. Notably, as in the survival analysis, cardiovascular diseases did not appear in the top features in either the multi-factorial models or the dementia type-specific models, with the exception of congestive heart failure for VaD. The absence of these comorbidities from the top features in our machine-learning models may suggest that cognitive decline is a stronger predictor of mortality in dementia patients than stroke or other comorbid cardiovascular conditions, though further studies could better interrogate this hypothesis.

Our study had several key strengths. First, the NACC database is the largest resource of its kind in the United States, covering a large, diverse patient population that was current through September 2021. Moreover, we highlight a conscious design choice in stratifying our data into train, internal test, and external test sets. By introducing a prospective external test set based on date, we were able to ascertain the ability of our models to predict mortality within a prospective cohort based on past data. By utilizing SHAP as our feature importance metric, we were able to gain valuable insight into the key factors underlying the predictions of our XGBoost models. This increased transparency also enables more informed decision-making, by facilitating the communication of results to clinicians and other non-technical stakeholders. Overall, the synergistic relationship between SHAP and machine-learning algorithms like XGBoost enhances the utility and scope of predictive machine learning. In our pan-dementia analysis, the use of two-feature and nine-feature (multi-factorial) models provided a parsimonious, clinically feasible framework for predicting dementia patient mortality, while in our sub-dementia analysis, the comparison of important predictors of mortality across various dementia types may help to guide precision management and treatment of dementia.

However, our study also had limitations. Due to the high prevalence of missing values, largely attributed to the difficulty in acquiring certain data (e.g., neuropathological data) and differences in clinical procedures across ADCs, many features were preliminarily eliminated. Moreover, many features within the NACC data measure similar phenomena, certain variables have changed over time as updates were made to the UDS form, and

many variables were derived from clinician diagnosis, precluding the use of a more granular feature selection method. By first eliminating variables with over 40% missing values and subsequently using MICE to impute the remaining features, we aimed to reduce some bias in the feature selection process[63], though we acknowledge the limitation of neglecting features that may only be ascertained by a selection of ADCs. We highlight that the best performance can be achieved if each ADC or clinic derives its own predictive model based on its respective available features. Moreover, our data-splitting method excluded patients who are lost to follow-up, which biases the deceased group towards a shorter survival time. This will likely make the predictors over different survival thresholds more similar to each other and overestimate the AUC values for the longer survival thresholds.

While XGBoost is known for its predictive power, the XGBoost-based dementia survival models have limitations that warrant further investigation. First, the models do not account for the dynamic nature of dementia progression and the potential for interventions to alter the course of the disease[64]. For instance, considering the impact of potential pharmacological treatments for dementia could help refine the survival prediction models. Moreover, current XGBoost models primarily rely on clinical and demographic data, which may not provide a comprehensive account of the complex biological mechanisms underlying dementia. Incorporating genetic and multi-omics data into the dementia survival models could enhance predictive performance. Future studies could explore the integration of data from various sources, such as genomics, transcriptomics, proteomics, and metabolomics, to create a more holistic understanding of dementia pathogenesis and identify novel biomarkers for prediction. The application of state-of-the-art deep learning architectures, such as transformers, may also improve predictive performance, especially with the incorporation of genetic and multi-omics data[65]. Deep-learning methods can capture even more complex, non-linear relationships and have demonstrated success in a wide range of biomedical applications, and may also improve prediction using longitudinal and multi-modal data.

Overall, this study revealed that machine-learning models have utility in predicting dementia patient mortality at various survival-time thresholds. Parsimonious models can be developed when limited clinical features are available, and dementia type-specific models can be used for distinguishing heterogeneous patient populations. If cross-validated and carefully implemented at the primary care level, such predictive models can improve personalized care of dementia.

## Data availability

The data used in this study can be requested from the National Alzheimer's Coordinating Center by completing the NACC data request form available at https://nacc.redcap.rit.uw.edu/surveys/?s=KHNPKLJW8TKAD4DA.

## Code availability

The code implemented in this study is available both in a GitHub repository at https://github.com/Huang-lab/dementia-survival-prediction and in a public repository[66] at https://doi.org/10.5281/zenodo.10392776.

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

## Acknowledgements

We would like to acknowledge the National Alzheimer's Coordinating Center and its participating patients and families who contributed data to the NACC database. The NACC database is funded by NIA/NIH Grant U24 AG072122. NACC data are contributed by the NIA-funded ADCs: P50 AG005131 (PI James Brewer, MD, PhD), P50 AG005133 (PI Oscar Lopez, MD), P50 AG005134 (PI Bradley Hyman, MD, PhD), P50 AG005136 (PI Thomas Grabowski, MD), P50 AG005138 (PI Mary Sano, PhD), P50 AG005142 (PI Helena Chui, MD), P50 AG005146 (PI Marilyn Albert, PhD), P50 AG005681 (PI John Morris, MD), P30 AG008017 (PI Jeffrey Kaye, MD), P30 AG008051 (PI Thomas Wisniewski, MD), P50 AG008702 (PI Scott Small, MD), P30 AG010124 (PI John Trojanowski, MD, PhD), P30 AG010129 (PI Charles DeCarli, MD), P30 AG010133 (PI Andrew Saykin, PsyD), P30 AG010161 (PI David Bennett, MD), P30 AG012300 (PI Roger Rosenberg, MD), P30 AG013846 (PI Neil Kowall, MD), P30 AG013854 (PI Robert Vassar, PhD), P50 AG016573 (PI Frank LaFerla, PhD), P50 AG016574 (PI Ronald Petersen, MD, PhD), P30 AG019610 (PI Eric Reiman, MD), P50 AG023501 (PI Bruce Miller, MD), P50 AG025688 (PI Allan Levey, MD, PhD), P30 AG028383 (PI Linda Van Eldik, PhD), P50 AG033514 (PI Sanjay Asthana, MD, FRCP), P30 AG035982 (PI Russell Swerdlow, MD), P50 AG047266 (PI Todd Golde, MD, PhD), P50 AG047270 (PI Stephen Strittmatter, MD, PhD), P50 AG047366 (PI Victor Henderson, MD, MS), P30 AG049638 (PI Suzanne Craft, PhD), P30 AG053760 (PI Henry Paulson, MD, PhD), P30 AG066546 (PI Sudha Seshadri, MD), P20 AG068024 (PI Erik Roberson, MD, PhD), P20 AG068053 (PI Marwan Sabbagh, MD), P20 AG068077 (PI Gary Rosenberg, MD), P20 AG068082 (PI Angela Jefferson, PhD), P30 AG072958 (PI Heather Whitson, MD), P30 AG072959 (PI James Leverenz, MD). This work was also supported by NIGMS R35GM138113 and ISMMS funds to K.H.

## Author contributions

K.H. conceived the research, and J.Z., L.S., and K.H. designed the analyses. Z.M. and K.C. provided the NACC data and consulted for its meaningful use. J.Z. and L.S. conducted the computational analyses. K.C. provided statistical suggestions. K.H. supervised the overall study. J.Z., L.S. wrote the manuscript, and K.H. edited the manuscript. All authors read, edited, and approved the manuscript.

## Competing interests

The authors declare no competing interests.
