## [Peer Review File · Communications Medicine]

Reviewers' comments:

Reviewer #1 (Remarks to the Author):

Zhang et al., present comprehensive machine learning modeling of patient mortality across multiple dementia types. Specifically, the authors utilized a dataset of 45,275 participants and 163,782 visit records from the U.S. National Alzheimer's Coordinating Center (NACC) to predict survival using various machine learning models. They achieved an AUC-ROC of 0.82 utilizing nine parsimonious for all one-, three-, five-, and ten-year thresholds after various feature selection. Interesting, unsupervised clustering of mortality predictors grouped vascular dementia with depression and Lewy body dementia with frontotemporal lobar dementia. Overall, this is an important and novel machine learning studies for dementia precision medicine. These machine models have potential application for future clinical studies and patient care of dementia.

The authors used XGBoost models encompassing all 189 features. The authors should provide rationale why selecting XGBoost models, not other types of machine learning models. In addition, it is unclear whether further feature selection can improve performance of prediction.

As shown in Tables 2 and 3, the authors may explain why ten-year models showed much better preference in comparison to one-year models, power issues or other explanations.

The authors are encouraged to discuss the limitations and future studies of current dementia survival models. For example, whether adding genetic or multi-omics data may improve the prediction power further in the future studies.

Several related studies may discussed, such as doi: [10.1038/s43587-021-00138-z](https://doi.org/10.1038/s43587-021-00138-z), and doi: [10.1016/j.celrep.2022.111717](https://doi.org/10.1016/j.celrep.2022.111717)

The authors may improve figure resolution and provide more discussion about the clinical interpretation of features listed in Figure 3.

Reviewer #2 (Remarks to the Author):

Brief summary of the manuscript

- The paper presented a study to predict mortality rate using different sets of clinical features across large populations with different dementia types.
- The authors use the survival model to identify patients with risks of near-term mortality at different time scales (from 1-10 years)
- The authors further use explainable machine learning models (XGBoost + TreeSHAP) to predict mortality at different time threshold to infer the clinical relevance of dementia subtypes. Two feature (age+global CDR) based XGBioost ML model with a stratified survival rate was first trained followed by a multivariable machine-learning model with pre-training feature selection and post-training feature importance analysis.
- The authors also use unsupervised clustering to group different dementia subtypes

Overall impression of the work

- The Survival analysis stratified by comorbidities is a strength.
- Investigating the difference among dementia subtypes in terms of clinical features and survival curve is an important research question with clinical interest, although some results seems to be missing in the submitted manuscript
- Class imbalance in the machine-learning model would be a big concern, and need to be addressed appropriately.
- The use of model explainable methods (SHAP) to evaluate feature importance is a strength, although

Specific comments

1. Investigating differences among dementia subtypes in terms of clinical features and survival curve is an important research question with clinical interest.

- [189] Using feature importance to cluster/group dementia subtypes is an interesting innovation of this study. The result of different feature importance profiles among different dementia subtypes is of great clinical interest.

- [253] Missing result figures. In the discussion, the authors mentioned that “Our dementia-type survival analysis confirmed this heterogeneity as survival probability differed drastically across groups of patients with different primary etiologic diagnoses.” It seems to be an important major finding in the study. However, I seems cannot find the corresponding figures in the submitted manuscript about survival analysis for different dementia subtypes, even in the supplementary material. Probably a figure is missing in the submitted version?

2. Handling the class imbalance in this study seems to be a major issue, and need to be handled properly with class-balanced data resampling or class-weighted loss function

- [Line 120/148] It’s natural that subjects at a later stage (e.g. 10 years from the initial visit) would have higher mortality rates. And if this is not adjusted, it’s hard to investigate whether the difference between a shorter survival threshold (e.g. 1-3 years) and longer (e.g. 10 years) is a genuine finding or simply due to the class imbalance issue.

- [180-188/Table 3]: year-5 threshold was selected because of its least class-unbalanced. However, after the population is stratified into dementia subtypes and non-demented populations, the survival/non-survival class would be unbalanced again, and that’s probably the reason for the results of higher AUC-PR for the demented group and higher AUC-ROC for the non-demented group. P.S. Table 3 seems to be incorrect, which is the same as table 2 in its current format.

3. The use of model explainable methods (SHAP) to evaluate feature importance is a strength.

- [164] It’s not clear in Fig 2b what the authors meant by “the direction of the effects began to reverse at the longer survival thresholds.” Is it among important variables or is it within variables? Which variables are affected, and by how much?

- [130] The criteria for the number (=9) of top SHAP-based features to select is not clear (Suppl Fig 4)

- The robustness of these top-selected features also need to be validated Either through bootstrapping or cross-validation.

4 [line 153 + suppl Fig 5] I appreciate the author’s efforts to evaluate the model performance among different ADRC sites, as that’s an important issue to address when analyzing large-scale multi-center data such as NACC. However, the fact that the sites are only stratified in the test set (which was wrongly called the validation set in the manuscript), means the data from the same site appears in both the training and the test set, therefore causing potential data leakage. If the author intend to evaluate the inter-site variability of the ML model, a more appropriate way would be to apply leave-site-out cross-validation, in which the nth site is left out for testing and the remaining n-1 sites are used for training. A proper discussion over this point would be appreciated.

Other Minor Comments to the manuscript content

- Table 2 / Supp Table 2 / line 114, etc.: the terminology of validation/test set seems to be used wrongly, validation set are used during the training to monitor and avoid overfitting, while the test set are independent external dataset to evaluate the model performance.
- Table 3 is incorrect, which currently the same copy of table 2.
- [Line 118] AUC-ROC/AUC-PR need to be introduced in full spell before using
- [Line 121] Shall be Supplementary Fig 2 rather than 3
- [Line 130] Shall be Supplementary Fig 3 rather than 4
- [145] sentence incomplete
- [427, 466] The paragraph about TreeSHAP in line 427 shall be part of the "Feature Importance" section in line [466].
- [Supplementary Table 2] doesn't properly separate the validation set and test set results (again, the use of these two terminology shall be switched as pointed out earlier), and the accuracy of one set of results seems to be missing

Reviewer #3 (Remarks to the Author):

Thanks for recommending me as a reviewer. In this paper, authors developed machine learning models predicting survival using a dataset of 45,275 unique participants and 163,782 visit records from the U.S. National Alzheimer's Coordinating Center (NACC). In this paper, the models achieved an AUC-ROC of over 0.82 utilizing nine parsimonious features for all one-, three-, five-, and ten-year thresholds. Overall, this study is well written. If the authors complete minor revisions, the quality of the study will be further improved.

1. line 109-120: In this study, the authors used XGB as an algorithm for ML. Why did the authors choose XGB among the boosting algorithms? For example, since there are more than 10,000 subjects, light GBM may perform better than XGB. If the author describes the reasons for choosing XGB in more detail, it can help readers understand.

2. line 466: Feature importance - In this study, the authors used SHAP as a method to analyze feature importance. However, the authors lack mention of SHAP in the results and discussion sections. If the authors expand the results and discussion sections, they can help readers understand.

3. Authors should add the study's limitations in the discussion section.

Point-by-point response to reviewers' comments:

We thank all three reviewers for taking their time and providing constructive reviews for our work. We addressed each of the suggestions made by the reviewers, which led to substantial improvement of the manuscript. Please find a point-by-point response to the reviewers' comments below:

Reviewer #1

Zhang et al., present comprehensive machine learning modeling of patient mortality across multiple dementia types. Specifically, the authors utilized a dataset of 45,275 participants and 163,782 visit records from the U.S. National Alzheimer's Coordinating Center (NACC) to predict survival using various machine learning models. They achieved an AUC-ROC of 0.82 utilizing nine parsimonious for all one-, three-, five-, and ten-year thresholds after various feature selection. Interesting, unsupervised clustering of mortality predictors grouped vascular dementia with depression and Lewy body dementia with frontotemporal lobar dementia. Overall, this is an important and novel machine learning studies for dementia precision medicine. These machine models have potential application for future clinical studies and patient care of dementia.

1. The authors used XGBoost models encompassing all 189 features. The authors should provide rationale why selecting XGBoost models, not other types of machine learning models. In addition, it is unclear whether further feature selection can improve performance of prediction.

RESPONSE: Thank you for this suggestion, and we agree that providing further rationale for selecting XGBoost as our algorithm of choice, as opposed to other high-performing boosting algorithms, would enhance our paper. To justify our choice of XGBoost, we have included a reference to an article comparing the accuracy and speed of various gradient boosting algorithms (doi: 10.1007/s10462-020-09896-5), which found that XGBoost provides an optimal balance between speed and accuracy, which is crucial given the size of our dataset and the associated computational complexity required for training and optimization. Moreover, we experimented with LightGBM, another boosting algorithm, and found that its performance was nearly identical to XGBoost (AUC-ROC score within 0.01 of one another) at all four survival thresholds, so we opted to proceed with XGBoost for our analyses. Accordingly, we have added the following paragraph summarizing our rationale for selecting XGBoost to our Methods:

“After experimenting with several machine learning algorithms (i.e., random forest, logistic regression, and gradient boosting), our machine learning algorithm of choice was eXtreme Gradient Boosting (XGBoost), a high-performance, tree-based ensemble learning method that uses gradient tree boosting to sequentially add new trees to reduce the errors from previous trees⁵⁶. As compared to other gradient boosting algorithms like light gradient-boosting machine (LightGBM), previous literature has found that XGBoost provides an optimal balance between accuracy and training speed⁵⁷. When we evaluated the predictive performance of XGBoost vs. LightGBM, we found that they achieved AUC-ROC scores within 0.01 of each other at all four survival thresholds: 0.81 vs. 0.82 for one-year, 0.82 vs 0.83 for

three-year, 0.82 vs. 0.83 for five-year, and 0.83 vs. 0.83 for ten-year. Thus, we opted to proceed with XGBoost for the rest of our analyses.”

As for further feature selection, we recognize that including more features may slightly increase predictive performance, but would come at the expense of parsimony, which drastically reduces clinical applicability. Ultimately, we opted to curate a small set of the most relevant features for dementia survival prediction that strikes a balance between predictive performance and clinical feasibility, and we have acknowledged this trade-off in our Results:

“Therefore, we derived a parsimonious and informative feature subset across all four survival time thresholds by training default XGBoost classifiers with five-fold cross validation on each of the four training sets and taking the union of the top n features from each model. Specifically, we selected n=5 as our cutoff as it provided considerable overlap among survival thresholds while still preserving enough features to be clinically informative (Supplementary Fig. 4b,c). Nonetheless, the selection criteria constitute a deliberate compromise, striking a balance between optimizing model performance and ensuring practicality for clinical applications.”

2. As shown in Tables 2 and 3, the authors may explain why ten-year models showed much better preference in comparison to one-year models, power issues or other explanations.

RESPONSE: Regarding the stronger predictive performance (particularly in terms of AUC-PR) of the five-year and ten-year models as compared to the one-year and three-year models, we believe that the distinction can be primarily attributed to higher class imbalance at the lower survival thresholds. This is exemplified by the almost linearly increasing AUC-PR scores from one-year up to ten-year, with each subsequently threshold demonstrating more mortality patients and fewer survival patients than the last. Even after adjusting the ‘scale_pos_weight’ XGBoost parameter to weigh mortality patients more heavily than survival patients, the one-year and three-year models still could not overcome the immense class imbalance, which would otherwise come at the expense of significantly poorer performance in other regards (e.g., AUC-ROC and accuracy). We have added an explanation of why the higher survival thresholds showed much better AUC-PR than the lower survival thresholds in our Results:

“All four models achieved an AUC-ROC of over 0.82, though the lower threshold models (i.e., one-year and three-year) struggled with respect to AUC-PR [...] Notably, model AUC-PR was worse in one/three-year survival but increased dramatically at higher survival time thresholds, which can likely be attributed to a higher proportion of mortality in patients and, thus, smaller class imbalances at these higher thresholds. Whereas the AUC-ROC metric can be optimistic on imbalanced classification problems, which is exemplified by the relatively consistency of AUC-ROC scores across all four survival thresholds, AUC-PR is more strongly affected by class imbalance, which is likely why AUC-PR scores improved so dramatically as the proportion of mortality patients increased.”

3. The authors are encouraged to discuss the limitations and future studies of current dementia survival models. For example, whether adding genetic or multi-omics data may improve the prediction power further in the future studies.

RESPONSE: Thank you for this suggestion, and we agree that further discussion of the limitations of current XGBoost dementia survival models would help to emphasize our study's strengths and shortcomings, as well as motivate future research. Accordingly, we have added a paragraph in our Discussion highlighting the limitations of current dementia survival models and suggestions for future research, which may include genetic or multi-omics data:

“While XGBoost is known for its predictive power, the XGBoost-based dementia survival models have limitations that warrant further investigation. First, the models do not account for the dynamic nature of dementia progression and the potential for interventions to alter the course of the disease. For instance, considering the impact of potential pharmacological treatments for dementia could help refine the survival prediction models. Moreover, current XGBoost models primarily rely on clinical and demographic data, which may not provide a comprehensive account of the complex biological mechanisms underlying dementia. Incorporating genetic and multi-omics data into the dementia survival models could enhance predictive performance. Future studies could explore the integration of data from various sources, such as genomics, transcriptomics, proteomics, and metabolomics, to create a more holistic understanding of dementia pathogenesis and identify novel biomarkers for prediction. The application of state-of-the-art deep learning architectures, such as transformers, may also improve predictive performance, especially with the incorporation of genetic and multi-omics data. Deep learning methods can capture even more complex, non-linear relationships and have demonstrated success in a wide range of biomedical applications, and may also improve prediction using longitudinal and multi-modal data.”

4. Several related studies may discussed, such as doi: 10.1038/s43587-021-00138-z, and doi: 10.1016/j.celrep.2022.111717

RESPONSE: Thank you for suggesting these related studies. We have added references to both papers in our Discussion.

5. The authors may improve figure resolution and provide more discussion about the clinical interpretation of features listed in Figure 3.

RESPONSE: We appreciate this suggestion, and we have incorporated additional clinical interpretation of the important features for each of the dementia type models, as well as for their associated clusters. To complement the clinical interpretation that we already have in our Discussion, we have added the following paragraph to the end of our Results:

“Overall, in the context of VaD and depression, mortality prediction was predominantly determined by age, body measurements, and vital signs, which is not surprising given that patients with VaD and/or depression often present strong physical deficiencies, in addition

to cognitive ones. Meanwhile, given that FTL and LBD are both associated with alterations in personality, behavioral changes, and motor symptoms, it is plausible that the model would group these conditions together. As for patients with AD or other forms of dementia, mortality prediction was determined almost entirely based on age and cognitive features (e.g., performance on cognitive screening tests). We postulate that the grouping of AD with other forms of dementia may be attributed to the National Institute on Aging and Alzheimer's Association's (NIA-AA) diagnostic criteria for Alzheimer's disease, which relies on a process of exclusion. Finally, the mortality prediction of individuals without dementia or with unknown dementia status was moderately influenced by age, biological sex, cognitive features, lifestyle factors (e.g., cigarette smoking), and geographical location. These findings align with previous research conducted on the general population."

Additionally, we have reproduced our figures at a resolution of 300 DPI.

Reviewer #2

Brief summary of the manuscript

- The paper presented a study to predict mortality rate using different sets of clinical features across large populations with different dementia types.
- The authors use the survival model to identify patients with risks of near-term mortality at different time scales (from 1-10 years)
- The authors further use explainable machine learning models (XGBoost + TreeSHAP) to predict mortality at different time threshold to infer the clinical relevance of dementia subtypes. Two feature (age+global CDR) based XGBoost ML model with a stratified survival rate was first trained followed by a multivariable machine-learning model with pre-training feature selection and post-training feature importance analysis.
- The authors also use unsupervised clustering to group different dementia subtypes

Overall impression of the work

- The Survival analysis stratified by comorbidities is a strength.
- Investigating the difference among dementia subtypes in terms of clinical features and survival curve is an important research question with clinical interest, although some results seems to be missing in the submitted manuscript
- Class imbalance in the machine-learning model would be a big concern, and need to be addressed appropriately.
- The use of model explainable methods (SHAP) to evaluate feature importance is a strength, although

Specific comments

1. Investigating differences among dementia subtypes in terms of clinical features and survival curve is an important research question with clinical interest.

- [189] Using feature importance to cluster/group dementia subtypes is an interesting innovation of this study. The result of different feature importance profiles among different dementia subtypes is of great clinical interest.

RESPONSE: Thank you.

- [253] Missing result figures. In the discussion, the authors mentioned that “Our dementia-type survival analysis confirmed this heterogeneity as survival probability differed drastically across groups of patients with different primary etiologic diagnoses.” It seems to be an important major finding in the study. However, I seems cannot find the corresponding figures in the submitted manuscript about survival analysis for different dementia subtypes, even in the supplementary material. Probably a figure is missing in the submitted version?

RESPONSE: The dementia-type survival analysis specifically corresponds to Figure 1b, which depicts the Kaplan-Meier estimator curve stratified by dementia type (i.e., primary etiologic diagnosis). The aforementioned comment in the Discussion regarding the heterogeneity in survival probability across different primary etiologic diagnoses is also delineated in the Results, and the corresponding figure is cited as follows:

“Survival probability differed across primary etiologic diagnoses of dementia types (Fig. 1b). Patients with prion disease showed less overall median survival time [...]”

We have also improved the resolution of our figures.

2. Handling the class imbalance in this study seems to be a major issue, and need to be handled properly with class-balanced data resampling or class-weighted loss function

- [Line 120/148] It’s natural that subjects at a later stage (e.g. 10 years from the initial visit) would have higher mortality rates. And if this is not adjusted, it’s hard to investigate whether the difference between a shorter survival threshold (e.g. 1-3 years) and longer (e.g. 10 years) is a genuine finding or simply due to the class imbalance issue.

- [180-188/Table 3]: year-5 threshold was selected because of its least class-unbalanced. However, after the population is stratified into dementia subtypes and non-demented populations, the survival/non-survival class would be unbalanced again, and that’s probably the reason for the results of higher AUC-PR for the demented group and higher AUC-ROC for the non-demented group. P.S. Table 3 seems to be incorrect, which is the same as table 2 in its current format.

RESPONSE: The issue of class imbalance is certainly a tricky and important one, and one that we grappled with extensively as we formulated the methodology for our study. Ultimately, we decided to use a class-weighted loss function (rather than a data resampling method) in the form of the ‘scale_pos_weight’ parameter for XGBoost, which we manually set to the ratio between the number of samples in the negative class (survival class) and the number of samples in the positive class (mortality class). In this way, the two classes are artificially balanced, without needing to directly manipulate the dataset via oversampling or undersampling. For instance, a ‘scale_pos_weight’ value > 1, as was the case for the one-year, three-year, and five-year survival thresholds, would result in greater penalization for the misclassification of mortality patients than for survival patients, and since we set the values of

'scale_pos_weight' manually, the penalization would be directly proportional to the magnitude of the imbalance. Moreover, a related study that used XGBoost on an Alzheimer's disease-focused gene set addressed imbalance by experimenting with both the 'scale_pos_weight' parameter and oversampling and found that the "weighted" method outperformed the "balanced" method (doi: 10.1038/s42003-022-03068-7). We also briefly experimented with SMOTE (Synthetic Minority Over-sampling Technique) on unoptimized XGBoost models and found that the performance was nearly identical to using 'scale_pos_weight,' while the oversampled datasets also took considerably longer to train on.

As this distinction is certainly an important one, we have added a section to our Methods entitled "Class Imbalance" in order to elaborate on our decision to use a class-weighted loss function (i.e., via 'scale_pos_weight') rather than a class-based resampling method:

"All four survival datasets exhibited some degree of class imbalance: the one-year, three-year, and five-year datasets contained a higher proportion of survival patients than mortality patients, whereas the ten-year dataset contained a higher proportion of mortality patients than survival patients. In order to address this class imbalance, we chose to apply a class-weighted loss function when training our models via XGBoost's built-in 'scale_pos_weight' parameter. This parameter controls the balance of positive and negative weights, such that setting this parameter equal to the ratio between the number of samples in the negative class and the number of samples in the positive class produces a class-weighted loss function for the XGBoost model. For the two-feature and multi-factorial models, the 'scale_pos_weight' parameter was set to 15.081 for one-year, 3.862 for three-year, 1.872 for five-year, and 0.476 for ten-year (Supplementary Table 4). We opted to use a class-weighted loss function rather than a class-based resampling method because previous studies have demonstrated superior performance of weighted methods as compared to resampling methods in addressing class imbalance, and we also wanted to avoid the additional computational complexity incurred by oversampling."

3. The use of model explainable methods (SHAP) to evaluate feature importance is a strength. - [164] It's not clear in Fig 2b what the authors meant by "the direction of the effects began to reverse at the longer survival thresholds." Is it among important variables or is it within variables? Which variables are affected, and by how much?

RESPONSE: We agree that the original wording of this sentence was unclear and possibly confusing. For better clarity, we have reworded the sentence to clearly specify the variables implicated in this "reversal" and the degree to which the relationship between these variables and mortality risk has been inverted:

"More years of smoking ('SMOKYRS') and more commission errors on the Trail Making Test Part B ('TRAILBRR') were also predictive of higher risk in one-year and three-year mortality. However, the inverse relationship was observed for five-year and ten-year mortality risk, as fewer years of smoking and fewer commission errors on the Trail Making Test Part B

became slightly more predictive of mortality than of survival at these longer survival thresholds.”

- [130] The criteria for the number (=9) of top SHAP-based features to select is not clear (Suppl Fig 4)

RESPONSE: Thank you for this comment, and we agree that we did not clearly explain how and why we selected the 9 features we used in our pan-dementia analysis. Specifically, we aimed to curate a set of features that demonstrated high predictive value across all four survival thresholds in order to avoid having a distinct feature set for each threshold, which curtails clinical feasibility as well. Thus, we chose to select the top n features from each survival threshold ranked by mean absolute SHAP value, as shown in Supplementary Fig. 4. We ultimately chose n=5 as our cutoff, as there was considerable overlap among the top 5 features across all four survival thresholds, while still retaining enough features to capture most of the variance in the dataset. To make this as clear as possible, we have updated Supplementary Fig. 4 to also include a line plot of the total number of features for varying cutoffs and a Venn diagram showcasing the number of overlapping features when n=5. We have also elaborated on our criteria for feature selection in our Results:

“Therefore, we derived a parsimonious and informative feature subset across all four survival time thresholds by taking the union of the top n features from each model. Specifically, we selected n=5 as our cutoff as it provided considerable overlap among survival thresholds while still preserving enough features to be clinically informative (Supplementary Fig. 4b,c). Nonetheless, the selection criteria constitute a deliberate compromise, striking a balance between optimizing model performance and ensuring practicality for clinical applications.”

- The robustness of these top-selected features also need to be validated Either through bootstrapping or cross-validation.

RESPONSE: Thank you for raising this concern. We did, in fact, use five-fold cross validation to generate these top-selected features. We first aggregated the SHAP values within each partition and then ranked the importance of each feature for each survival threshold’s model, before taking the union of the top 5 features from each model. Moreover, the presence of multiple overlapping top features between survival thresholds provides some reassurance regarding the robustness of the selected features. However, for sake of clarity, we have added the following line to the Feature Selection section of our Methods:

“In our study, we trained default XGBoost classifiers with five-fold cross-validation on each of the four training sets, aggregating SHAP values across each partition before taking the union of the top five features from each model, ranked in order of decreasing mean absolute SHAP value.”

We have also added a similar comment in our Results:

“Therefore, we derived a parsimonious and informative feature subset across all four survival time thresholds by training default XGBoost classifiers with five-fold cross validation on each of the four training sets and taking the union of the top n features from each model.”

4 [line 153 + suppl Fig 5] I appreciate the author’s efforts to evaluate the model performance among different ADRC sites, as that’s an important issue to address when analyzing large-scale multi-center data such as NACC. However, the fact that the sites are only stratified in the test set (which was wrongly called the validation set in the manuscript), means the data from the same site appears in both the training and the test set, therefore causing potential data leakage. If the author intend to evaluate the inter-site variability of the ML model, a more appropriate way would be to apply leave-site-out cross-validation, in which the nth site is left out for testing and the remaining n-1 sites are used for training. A proper discussion over this point would be appreciated.

RESPONSE: We thank the reviewer for catching this potential oversight, as we did not account for data leakage when evaluating model performance across different ADC sites. As the reviewer suggested, we have revised our analysis of inter-site variability by performing a leave-site-out cross-validation, using all but one site to train the XGBoost model and subsequently testing on the remaining site that was not already used for training. Evidently, the leave-site-out cross-validation did not substantively change our findings, as model performance still remained relatively consistent across ADCs with sufficiently large patient populations. We have updated the plots in Supplementary Fig. 5 and added a description of the leave-site-out cross-validation in our Results:

“To avoid potential data leakage between sites, we performed leave-one-(site)-out cross-validation (LOOCV), in which training data from all but one site was used for training, while the internal test data from the leftover site was used for testing. This way, each ADC was tested and evaluated using models that were trained on data exclusively from other ADCs. Overall, model performance remained consistent across ADCs, with discrepancies primarily occurring in the ADCs with the smallest patient populations. The full AUC-ROC curves stratified by ADC are available in Supplementary Fig. 5 and demonstrate the broad generalizability of our model.”

Other Minor Comments to the manuscript content

- Table 2 / Supp Table 2 / line 114, etc.: the terminology of validation/test set seems to be used wrongly, validation set are used during the training to monitor and avoid overfitting, while the test set are independent external dataset to evaluate the model performance.

RESPONSE: We agree that we may have been conflating the terminology in our usage of the term “validation set,” which was, as the reviewer mentioned, an external test set for evaluating model performance as opposed to a validation set for optimizing parameters during training. To avoid potential misunderstanding, we have changed all instances of “test set” to “internal test

set” and “external validation set” to “external test set” in both the manuscript and the figures, which more accurately reflects our data splitting workflow. We thank the reviewer for helping us adopt the more standardized terminology to improve clarity.

- Table 3 is incorrect, which currently the same copy of table 2.

RESPONSE: Thank you for catching this. We have replaced the duplicate of Table 2 with the correct version of Table 3, which contains the predictive performance metrics of the individual dementia type models, in the revised manuscript.

- [Line 118] AUC-ROC/AUC-PR need to be introduced in full spell before using

RESPONSE: Thank you for this suggestion. We have included the full spellings of AUC-ROC and AUC-PR when we first use them:

“The two-feature XGBoost models achieved an AUC-ROC (area under the receiver operating characteristic curve) of over 0.76 at all survival thresholds, though the higher thresholds achieved much higher AUC-PR (area under the precision-recall curve) scores...”

- [Line 121] Shall be Supplementary Fig 2 rather than 3

RESPONSE: Supplementary Fig. 2 corresponds to the flowchart of data splitting, whereas Supplementary Fig. 3 corresponds to the ROC curves of the two-feature models. In this case, the latter was what we intended, as we stated that “[...] the AUC-ROC curves are shown in Supplementary Fig. 3”:

- [Line 130] Shall be Supplementary Fig 3 rather than 4

RESPONSE: Supplementary Fig 3. corresponds to the ROC curves of the two-feature models, whereas Supplementary Fig. 4 corresponds to the feature selection plots. In this case, the latter was what we intended, as we stated that “[...] much of the explainability of the predictions could be attributed to these top few features alone”, in reference to the feature selection bar plots.

- [145] sentence incomplete

RESPONSE: Thank you for catching this error. We have updated the sentence as follows:

“All four models achieved an AUC-ROC of over 0.82, though the lower threshold models (i.e., one-year and three-year) struggled with respect to AUC-PR.”

- [427, 466] The paragraph about TreeSHAP in line 427 shall be part of the “Feature Importance” section in line [466].

RESPONSE: Thank you for the suggestion. We have moved the sentence about TreeSHAP to the beginning of the “Feature Importance” section.

- [Supplementary Table 2] doesn't properly separate the validation set and test set results (again, the use of these two terminology shall be switched as pointed out earlier), and the accuracy of one set of results seems to be missing

RESPONSE: Thank you for catching this error. We have corrected Supplementary Table 2, which now includes the complete performance metrics from both the internal test set (formerly test set) and external test set (formerly validation set).

Reviewer #3

Thanks for recommending me as a reviewer. In this paper, authors developed machine learning models predicting survival using a dataset of 45,275 unique participants and 163,782 visit records from the U.S. National Alzheimer's Coordinating Center (NACC). In this paper, the models achieved an AUC-ROC of over 0.82 utilizing nine parsimonious features for all one-, three-, five-, and ten-year thresholds. Overall, this study is well written. If the authors complete minor revisions, the quality of the study will be further improved.

1.line 109-120: In this study, the authors used XGB as an algorithm for ML. Why did the authors choose XGB among the boosting algorithms? For example, since there are more than 10,000 subjects, light GBM may perform better than XGB. If the author describes the reasons for choosing XGB in more detail, it can help readers understand.

RESPONSE: Thank you for this suggestion. To justify our choice of XGBoost as opposed to other boosting algorithms such as LightGBM, we have referenced an article comparing the accuracy and speed of various gradient boosting algorithms (doi: 10.1007/s10462-020-09896-5), which found that LightGBM is the fastest but not the most accurate, whereas XGBoost provides an optimal balance between speed and accuracy, which is crucial given the size of our dataset and the associated computational complexity required for training and optimization. Moreover, we experimented with LightGBM and found that its performance was nearly identical to XGBoost (AUC-ROC score within 0.01 of one another) at all four survival thresholds, so we opted to proceed with XGBoost as our algorithm of choice. However, to make this clear to the reader, we have added the following paragraph to our Methods:

“After experimenting with several machine learning algorithms (i.e., random forest, logistic regression, and gradient boosting), our machine learning algorithm of choice was eXtreme Gradient Boosting (XGBoost), a high-performance, tree-based ensemble learning method that uses gradient tree boosting to sequentially add new trees to reduce the errors from previous trees⁵⁶. As compared to other gradient boosting algorithms like light gradient-boosting machine (LightGBM), previous literature has found that XGBoost provides an optimal balance between accuracy and training speed⁵⁷. When we evaluated the predictive performance of XGBoost vs. LightGBM, we found that they achieved AUC-ROC scores within

0.01 of each other at all four survival thresholds: 0.81 vs. 0.82 for one-year, 0.82 vs 0.83 for three-year, 0.82 vs. 0.83 for five-year, and 0.83 vs. 0.83 for ten-year. Thus, we opted to proceed with XGBoost for the rest of our analyses.”

2.line 466: Feature importance - In this study, the authors used SHAP as a method to analyze feature importance. However, the authors lack mention of SHAP in the results and discussion sections. If the authors expand the results and discussion sections, they can help readers understand.

RESPONSE: We agree that further discussion the SHAP method may give readers greater insight into its utility for clinical interpretability. Thus, we have expanded the Results section to include a more in-depth description of the SHAP method and its relevant use cases:

“We then used SHapley Additive eXplanations (SHAP), a robust, game-theoretic framework for explaining model output, to identify the important features within each survival threshold. These initial results identified numerous recurring features among the top features, ranked by SHAP values.”

We have also added a comment about the SHAP method’s strengths in regard to ease of interpretation of directional impact:

“Another key strength of the SHAP method is its ability to determine not only the magnitude of a feature’s effect on model prediction, but also the direction of its effect.”

Additionally, we have elaborated on our choice of SHAP as a feature selection and feature importance metric in our Discussion:

“By utilizing SHAP as our feature importance metric, we were able to gain valuable insight into the key factors underlying the predictions of our XGBoost models. This increased transparency also enables more informed decision-making, by facilitating the communication of results to clinicians and other non-technical stakeholders. Overall, the synergistic relationship between SHAP and machine learning algorithms like XGBoost enhances the utility and scope of predictive machine learning.”

3. Authors should add the study's limitations in the discussion section.

RESPONSE: Thank you for this suggestion. We have already included a brief paragraph summarizing our study’s main limitations with regard to data preprocessing, but we agree that more focus could be placed on limitations in other aspects of our study. Accordingly, we have added an additional paragraph in our Discussion highlighting the limitations of XGBoost as our algorithm of choice, along with suggestions for future research, which may opt to use deep learning methods and/or incorporate various multi-omics data:

“While XGBoost is known for its predictive power, the XGBoost-based dementia survival models have limitations that warrant further investigation. First, the models do not account for the dynamic nature of dementia progression and the potential for interventions to alter the course of the disease. For instance, considering the impact of potential pharmacological treatments for dementia could help refine the survival prediction models. Moreover, current XGBoost models primarily rely on clinical and demographic data, which may not provide a comprehensive account of the complex biological mechanisms underlying dementia. Incorporating genetic and multi-omics data into the dementia survival models could enhance predictive performance. Future studies could explore the integration of data from various sources, such as genomics, transcriptomics, proteomics, and metabolomics, to create a more holistic understanding of dementia pathogenesis and identify novel biomarkers for prediction. The application of state-of-the-art deep learning architectures, such as transformers, may also improve predictive performance, especially with the incorporation of genetic and multi-omics data. Deep learning methods can capture even more complex, non-linear relationships and have demonstrated success in a wide range of biomedical applications, and may also improve prediction using longitudinal and multi-modal data.”

REVIEWERS' COMMENTS:

Reviewer #1 (Remarks to the Author):

The authors have addressed my comments well.

Reviewer #2 (Remarks to the Author):

I appreciate the authors' revision to the manuscript, and addressed all my comments and concerns in previous round of review.

I have one remaining suggestion: the results from Figure 2b about reversed feature importance for the ('SMOKYRS') and ('TRAILBRR') from shorter years' threshold to longer year's threshold is interesting and potentially of great clinical interest. I'd suggest the authors to elaborate on the implication of the results more in the discussion section.

Reviewer #3 (Remarks to the Author):

The authors have faithfully completed the revision.

Reviewer #2

I appreciate the authors' revision to the manuscript, and addressed all my comments and concerns in previous round of review. I have one remaining suggestion: the results from Figure 2b about reversed feature importance for the ('SMOKYRS') and ('TRAILBRR') from shorter years' threshold to longer year's threshold is interesting and potentially of great clinical interest. I'd suggest the authors to elaborate on the implication of the results more in the discussion session.

RESPONSE: Thank you for this suggestion, and we agree that this finding is, indeed, of potentially great clinical interest. We have added the following sentences to our Discussion, highlighting some potential explanations for the reversed feature importance of 'SMOKYRS' and 'TRAILBRR' at the longer survival thresholds:

“Interestingly, at the longer survival thresholds, more years of smoking and more commission errors on the Trail Making Test Part B became more associated with survival than with mortality. Some possible explanations for these trends are that any protective effects of smoking cessation (which was not captured by the 'SMOKYRS' variable) against dementia risk and mortality might only be realized in long-term outcomes⁴³ and that the amount of time required to complete the Trail Making Test Part B (captured by 'TRAILB') might be a more consistent performance metric for predicting long-term mortality than the number of commission errors made (captured by 'TRAILBRR'). However, it is also important to note that the data for the longer survival thresholds included a slightly different subset of patients (i.e., a higher proportion of patients from the early 2010s) due to the more years of follow-up data required, so future studies might seek to further investigate these hypotheses.”